# The challenges of transgender and nonbinary graduate students in chemistry: A qualitative study on trans identity, science culture, and institutional support using reflexive thematic analysis

**Michelle M. Nolan**[1]*, **Isaac M. Blythe**[2], **Paulette Vincent-Ruz**[3]*

**1** Marston Science Library, University of Florida, Gainesville, Florida, United States of America,
**2** Department of Chemistry, Carleton College, Northfield, Minnesota, United States of America,
**3** Department of Chemistry & Biochemistry, New Mexico State University, Las Cruces, New Mexico, United States of America

* michellenolan@ufl.edu (MMN); pvr@nmsu.edu (PVR)

## Abstract

Transgender, nonbinary, two spirit, and gender-expansive students (herein *trans students*) are marginalized in higher education and have significantly different college experiences than their cisgender peers. Using in-depth interviews modeled after Sista Circles methodology and applying reflexive thematic analysis, this qualitative research illuminates the nuanced experiences of trans students navigating chemistry PhD programs ($N = 10$). The participants' counterstories revealed tensions between their identities as trans people and their identities as chemists, where STEM professional culture encouraged the participants to cover and separate their transness from their graduate education. The data demonstrated that these students navigated a complicated process when choosing a graduate program and deciding whether to share their trans identities in their institutions. Participants also encountered cisnormative institutional structures, including program applications and information technology systems, which enforced usage of their legal name and gender marker data in the academy. These results highlight disparities between institutional rhetoric regarding LGBTQ+ inclusion and tangible support for trans graduate students. From grappling with the absence of supportive policies to advocating for institutional change, participants confronted systemic barriers that impeded their academic and personal growth. This study underscores the imperative for transparent and proactive support structures within STEM academic departments to foster an environment where trans individuals can thrive.

## Introduction

Transgender, nonbinary, two spirit, and gender-expansive students (herein referred to by the umbrella term *trans* students) experience marginalization in higher education and have significantly different college experiences than their cisgender peers [1–4]. Furthermore,

**Data availability statement:** All relevant data are within the Paper and its Supporting Information files.

**Funding:** The author(s) received no specific funding for this work.

**Competing interests:** The authors have declared that no competing interests exist.

science, technology, engineering, and mathematics (STEM) academic communities, including chemistry, are known to be less diverse than other disciplines due to oppressive learning and working environments, especially along dimensions of gender, race, and disability [5–10]. However, few works have examined the intersection of scientific culture with LGBTQ+ identity. We know from a handful of scholarly reports and testimonies in popular media that chemistry is hostile to LGBTQ+ people of all professional ranks [11–14], with trans chemists reporting more encounters with exclusionary workplaces than cisgender chemists or LGBTQ+ physical scientists in other disciplines [15]. These hostile learning and working environments ultimately harm trans scientists' wellbeing and drive them out of the scientific field, with trans scientists experiencing worse health outcomes and being more likely to leave STEM than cisgender people [8,15]. Confronting hostile discipline cultures and empowering trans students in STEM is a fundamental matter of human dignity, right to education, and right to employment.

Is therefore important for researchers to examine the STEM trajectories of trans scientists at different career stages. For example, entry into graduate school is a pivotal career moment for students seeking advanced degrees in their chosen fields. However, beginning doctoral studies also involves navigating opaque expectations, unspoken norms, and complex power dynamics [16–18]. For trans students, this period carries additional complicating factors, such as contending with cisnormative academic spaces, negotiating visibility and safety, acclimating to chilly professional cultures, and seeking affirming mentors within their field. These intersecting pressures make the graduate school experience a crucial and uniquely complex time for trans students, shaping not only their academic journey but also their sense of agency, "outness," and safety. In a recent study, a nonbinary chemistry undergraduate student said they "wanted to enjoy chemistry more and loved what they were learning," but decided to leave the field after graduation because of an unwelcoming professional culture that made them "feel like I'm not supposed to be here" [19]. Other studies have shown that at the graduate education level, trans students face chronic misgendering in the natural sciences and describe STEM departments as "not LGBTQ-informed or savvy," forcing them to accept trans-hostile climates in order to obtain advanced degrees [20,21]. These reports signal that the transition from undergraduate education to graduate school is an important career juncture for trans students, during which they must decide whether staying in STEM is worth experiencing marginalization.

However, most published work aiming to characterize or ameliorate systemic oppression in STEM education thus far has focused on marginalized identities that are easily measurable. Because trans identities are not obviously apparent and must be disclosed by individual people, trans people are made invisible or deliberately excluded from research data collection, and they are overlooked in discussions of gendered oppression that center cisgender women [22–25]. Little data exist on the number of trans people in STEM education; the United States National Center for Science and Education Statistics does not currently collect data on LGBTQ+ identities, though they plan to begin doing so in 2025 after LGBTQ+ community pressure [26]. However, quantitative endpoint data will not be useful without understanding the experiences and barriers trans students face because quantitative methods cannot fully capture the experiences of trans people [22]. The reliability of quantitative data can be improved by triangulating with qualitative research that amplifies the perspectives of trans students in STEM programs, providing insights to the root causes behind measurable statistics (e.g., degree completion rate). For example, the collection of counterstories is one method that allows members of marginalized communities to express how systemic factors impact their experiences [27,28].

This work presents the results from a research study that co-created trans-informed counterstories about the graduate application process and subsequent lived experience in

chemistry graduate departments. Using qualitative methodologies to highlight trans graduate students' testimonials, we explored trans students' decision-making processes while beginning doctoral programs in chemistry and uncovered systemic barriers that they encountered. In addition to bringing attention to the tangible barriers to educational access participants of this study faced, we pose that there are nuances to the culture of chemistry that inhibit trans students' ability to present their full selves and impact trans students' success in their graduate programs.

## Theoretical framework

We ground the present work in the call by Jourian and Nicolazzo to bring our communities to the research table and develop knowledge *with* trans people, rather than *on* or *for* trans people, in a collective movement towards trans liberation [29]. We aimed to work in collaboration with trans students in chemistry to collectively build counterstories [27]. The goal was to surface evidence of barriers that specifically affect trans students so institutions and chemistry departments can address them. We reject the characterization of trans people as broken or abnormal, instead turning the focus of our interrogation on the systems that enforce cisnormativity and produce material harm. Our work instigates an intervention not at the personal level (e.g., mentorship, counseling), but at the structural level through challenging harmful institutional policies and cultures. Our research team brings together expertise in chemistry education research, trans community organizing, and lived experience with the subject matter: combining theory with an analysis of material conditions allowed us to stay focused on how cisnormative systems, as Stryker describes, are "never mere abstractions; they are systems of power that operate on actual bodies, capable of producing pain and pleasure, health and sickness, punishment and reward, life and death" (Stryker 2006). We center and amplify the experiences of trans graduate students to make an urgent call for transformative change in graduate education during a time of rapidly intensifying trans antagonism.

This research philosophy also is present in our theoretical framework (Fig 1) which follows an ecological structure. The ecological structure allows us to focus our discussion on systemic factors. The following overview is not to present an exhaustive literature review about trans students or to describe all systems that subjugate trans people, but rather to draw attention to

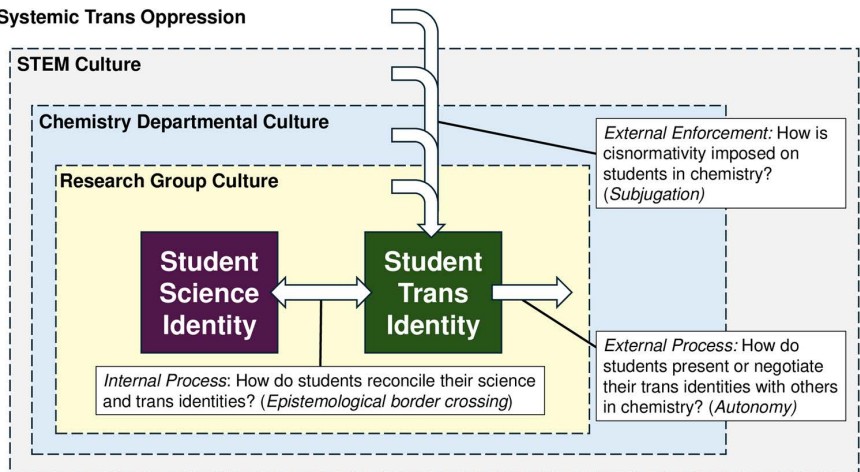

**Fig 1. Theoretical framework for the present study.**

elements of chemistry as an academic discipline that contextualize the counterstories of our participants.

## Epistemological border crossing

Epistemological border crossing refers to the tension marginalized students experience while trying to reconcile their science identities (i.e., personal identification and recognition by others as a scientist) with the social identities that they consider fundamental to their being [30]. When science and social identities are difficult to reconcile, marginalized scientists struggle to thrive in their chosen STEM trajectories. Epistemological border crossing emphasizes how internalized narratives about how a scientist looks/acts, encounters with oppressive professional cultures, and experiences of being othered in STEM spaces hinders students' abilities to "cross borders" or fully integrate their identities.

We pose that for trans students, the epistemological border crossing process will be facilitated or impeded by the influences of others during graduate school as students navigate the external processes of trans identity presentation and negotiation in their departments. This proposition is in accordance with what Mattheis, Cruz-Ramírez De Arellano, and Yoder proposed in their model of *queer STEM identity* [31]. In their work, the researchers proposed three overlapping processes for queer STEM identity development, each mediated by different internal and external factors: 1) *Defining* of a queer identity, or how individuals came to understand themselves as queer in terms of gender and/or sexuality; 2) *Forming* of a specific STEM identity related to their field of work; and 3) *Navigating* expression of identity in places of work or study. We conceptualized these decisions about defining one's trans identity through a frame of *autonomy*, wherein students made decisions about how to share or withhold information about their transness according to their own assessments and desires in a given situation. Internal epistemological border crossing (*Forming* of a specific STEM identity) is impeded by the external enforcement of cisnormativity, which is imposed on trans students from different ecological systems influencing their graduate studies (their research groups, their departments, their institutions, STEM culture, state governments, and so on), which we have conceptualized as *subjugation*. Subjugating forces impose gender binarism, cissexism, and other forms of antagonism on trans students, encouraging identity compartmentalization (*Navigating* expression of identity).

## Systemic trans oppression

Subjugation of trans people is pervasive and insidious. In the setting of higher education, a growing body of research demonstrates the harmful impacts of cisnormative collegiate settings on trans undergraduate students, including complicated daily negotiations regarding outness [1,32], bodily safety on campus [2], access to gendered spaces [33,34], hostile social climates [35], hostile STEM classrooms [36,37], mental health challenges [38], exposure to microaggressions [39], and much more. Fully describing the oppression of trans students is outside the scope of this article, but we point readers towards review articles [40–42] and longform texts [1,43] for in depth context. In contrast, research describing the experiences of trans graduate students remains slim, even though graduate students occupy a very different position in the academy than undergraduates. In particular, Goldberg and coauthors have led the consideration of trans graduate students, examining their unique stressors in graduate programs [44], decision-making process for choosing doctoral programs [21], and considerations moving from graduate school into their careers [45]. Recognizing that trans graduate students have unique needs that have not received adequate attention, Knutson published a call to action reflecting on what academics can do

to make graduate programs more welcoming for trans students, emphasizing the need for "broad and extensive" improvements [46].

## Stem culture

Systemic oppression of LGBTQ + people continues to thrive in STEM fields in part because STEM professionals falsely define their knowledge domains as purely technical, apolitical, asocial, and meritocratic, rendering scientists' personal identities as irrelevant to their scientific work and shielding oppressive conditions from challenge [47,48]. For example, the professional culture of engineering was characterized by Cech as having an "ideology of depoliticization," or a belief that engineering should be disconnected from discussions of social issues that "taint" engineering as a pure, objective pursuit [47]. Trans identity itself is not inherently political, but is increasingly politicized by a reactionary "culture war" narrative that seeks to deny trans livability [49–53]. By the logic of depoliticized STEM ideologies, transness itself is therefore interpreted as irrelevant or incompatible with STEM. In practice, the proclaimed neutrality of STEM enforces a cisheteronormative culture that has been likened to "don't ask, don't tell," where LGBTQ+ scientists are made invisible or actively excluded [31,54–58]. Disclosure of LGBTQ+ identity is perceived as unprofessional and socially inappropriate in science spaces, making daily interactions a constant source of emotional labor for trans students [36,58,59]. A pervasive "dude culture" in STEM departments assumes students are cisgender and heterosexual men, prizing hypermasculinity and thereby excluding or antagonizing LGBTQ + students who do not fit [60,61]. Ultimately, students are socialized into reproducing trans oppression as part of the process of becoming a STEM professional through official and unofficial pedagogical encounters.

## Departmental and research group culture

As simultaneous students, workers, and instructors, the material conditions of STEM graduate students' daily lives are distinctly different from those of undergraduate students on the same campuses and are variable to their specific fields. For example, doctoral advisors hold significant academic, financial, and social power over graduate students, making advisor-mentee relationships precarious for trans students with regards to presenting their trans identities when there is fear of rejection or retaliation [44,46]. Quality of doctoral mentorship has been well linked with graduate student success, but queer graduate students in STEM report strained relationships with advisors [62]. Some studies show that graduate students have an improved sense of belonging working with advisors who share their identities, including that women STEM students prioritize choosing women advisors over their research agendas as a strategy to minimize gendered discrimination [63]. However, since STEM faculty themselves are pressured to cover and hide their LGBTQ + identities [56,64], finding an advisor with shared queer identities may be precluded for trans students.

Furthermore, graduate education in chemistry is heavily focused on the practice of conducting laboratory work for long hours. Many graduate students have reported being explicitly or implicitly discouraged from participating in activities outside of the laboratory because they were characterized as distractions from their scientific work, including joining identity-based student organizations or spending time with other marginalized students outside their research group [65]. The insular nature of research labs, which one chemistry student described as "factions" where students were "sequestered with their group," kept students oppressive silos rife with microaggressions and inhibited the development of social support networks with others. Because trans students must constantly assess the people around them and negotiate the presentation of their trans identities to others, navigating research groups

may be a particularly challenging and unavoidable site of marginalization, with even further exposure to racialized transphobia for trans students of color. The isolation enforced by research group dynamics may also prevent trans graduate students from building the kinds of queer community spaces that undergraduate STEM students create to support one another amidst hostile educational cultures [66].

## Methodology

### Study design

A central goal in our study design was to provide a space where trans chemists felt like they were active co-constructors of knowledge and research seeking the liberation of trans people. This ethos is consistent with what other trans scholars have done to continue to push research in the field as something "done for the community" rather than "done to them" [29,67]. Trans scholars have pushed for the development of transfeminist methodologies and trans epistemologies, both of which prioritize practices that lead to liberation, the center trans people's unique experiences, and affirm the notion that trans people are not broken and rather just as Nicolazzo grapples a trans epistemology where trans people "are all that exists, all that there is" [68].

With this motivation, we found inspiration in Sista Circles methodology for the basis of our study. Sista Circles Methodology (SCM) is a Black feminist technique that fosters the collective building of counterstories by centering participant empowerment [69–72]. In SCM, participants actively engage as co-constructors in the research process and all research facilitators must be members of the community, mitigating power dynamics and challenging researcher-subject extractive dynamics [73,74]. It is crucial to emphasize that SCM shouldn't be misconstrued as merely a different name for a focus group. A central aspect of Sista Circles is a focus on participant empowerment in community, which is achieved through providing space outside the white gaze and encouraging the ways that Black women "share experiences, knowledge, wisdom, and power" to strengthen one another [73]. This intention fundamentally changes the dynamic from just an extractive data collection process to one where participants are actively working together in deconstructing oppressive experiences [75]. Furthermore, one of the central pillars of Sista Circles is the centering of Black women's unique forms of communication: Sista Circles are places where ways of expression, including the blended use of Black English vernacular, mainstream American vernacular, and nonverbal communication, are not policed or judged [73]. As non-Black researchers, the conscious choice of employing a Black feminist methodology as the basis for our study is a nuanced one. We recognize the uniqueness of SCM to serve Black women's liberation and we honor their contribution to liberatory praxis by being mindful of how we applied it to this study.

Reflecting on SCM as a methodology allowed us to provide participants with a space to collectively build stories as a means of both healing and creating paths towards trans liberation regarding how institutions must change to allow them to thrive. In this space, participants could freely use trans community language without having to explain its meaning to cis spectators, including the insider slang and modes of expression (e.g., sarcasm) that naturally arise when trans people are together. The rationale of co-construction guided our use of a semi-structured interview protocol, encompassing starting questions vital to our research objectives while allowing participants to lead discussions towards the topics they found most important to interrogate as a group.

The participants also responded to a brief questionnaire at the conclusion of the group interviews (S1 Appendix.) with information about the pseudonyms they chose for the purposes of the study, self-described pronouns, self-described gender identities, and approximate

institution size (Table 1). We want to acknowledge that gender identity does not exist in isolation: trans people conceptualize their genders through the lenses of race, ethnicity, and other salient identities [76–79]. Trans people who are multiply marginalized face unique barriers, especially for trans women of color who face the most violent enforcement of trans subjugation [80,81]. While we would have liked to collect other demographic information, including the participants' racial and ethnic identities, the research team felt the risk of participant exposure would be too high due to the very small population of "out" trans graduate students in the field of chemistry.

## Study participants eligibility and exclusion criteria

Participants were required to meet two eligibility criteria:

1. Be current doctoral students in chemistry departments who had not yet defended their PhD dissertations.

2. Self-identify as part of the trans community.

The trans student experience varies in levels of "outness" in their personal and professional lives. Furthermore, gender identities are not fixed or immutable. Therefore, any student that was currently gender-questioning or not "out" in a professional setting was eligible to participate. The team intentionally used an expansive definition of "trans" in the recruitment process, welcoming a wide range of gender identities and expressions as communicated through the following recruitment statement:

> *"We use the term 'trans' as a broad umbrella for all people whose gender identity and/or gender expression differs from what is culturally assigned, including (but not limited to) identities like transgender, nonbinary, genderqueer, agender, gender fluid, multigender, two spirit, and beyond. We welcome transfolk to participate regardless of how out, stealth, or currently gender-questioning they are in their personal and/or professional lives."*

Participants were recruited by circulating an interest form in Qualtrics through chemistry-specific LGBTQ+ email lists.

## Group interview procedure

Prior to the group interviews, participants were instructed to choose pseudonyms for themselves to use during the study. The rationale behind requesting participants choose their own

**Table 1. Profiles of Study Participants.**

| Interview Group | Pseudonym | Self-Described Pronouns | Self-Described Gender | Graduate Institution Enrollment Size |
|---|---|---|---|---|
| Group 1 | Cameron | they | nonbinary | Large (more than 30,000 students) |
| Group 1 | Jack | he or they | transmasc | Large (more than 30,000 students) |
| Group 1 | Kayden | they | nonbinary, genderfluid, transmasc | Small (fewer than 5,000 students) |
| Group 2 | Anna | she | trans female | Medium (15,001–30,000 students) |
| Group 2 | Eris | they | nonbinary, transfemme | Haven't chosen a school yet |
| Group 2 | Nat | they or she | nonbinary, transfemme | Large (more than 30,000 students) |
| Group 3 | Alex | she or they | transfemme, genderfluid | Small (fewer than 5,000 students) |
| Group 3 | Farren | he or they | trans man | Large (more than 30,000 students) |
| Group 3 | Indigo | she | trans woman | Medium (15,001–30,000 students) |
| Group 3 | Theo | she or sie/hir/hirs | genderfluid, genderqueer | Large (more than 30,000 students) |

pseudonyms included the following considerations: 1) The use of pseudonyms protected the privacy of participants from one another and from the research team during the group interview; 2) We were able to avoid the dehumanizing framing posed by participant numbers, enabling participants to refer to one another with a pseudonym organically during conversations; 3) We affirmed trans people's agency, right to self-description, and personal expression by allowing choice; 4) The research team avoided projecting our assumptions about participants by choosing pseudonyms on their behalf. Participants engaged in group discussions identified only by their chosen pseudonyms.

The interviews were conducted online via Zoom and recorded with participant consent. Audio was the only source used for the purposes of transcription and any potentially identifying information was redacted from the transcripts before analysis. Group sizes ranged from 3 to 4 participants with a total of 10 participants divided into three sessions, each lasting 90 minutes. Participants were not required to turn their Zoom cameras on, but most elected to share their video anyway. We paid special attention to observe when participants reacted to other speakers (e.g., nodded, snapped fingers, rolled eyes) and we encouraged deeper discussion in these moments.

To foster an environment where participants felt comfortable sharing their stories, the data collection process was solely facilitated by members of the research team who identified as trans, promoting a space of shared lived experience. The two facilitators shared personal stories with the group to make the conversation reciprocal and challenge the researcher/subject extractive dynamic, building trust with the participants through demonstrating our "insider" connection to the subject matter. Of note, we affirmed to participants that they did not need to define community terminology or justify why their viewpoints are important, asking them to speak plainly as they would with other trans folks in their daily lives. Generally, the facilitators offered starting prompts for discussion, the participants shared their initial perspectives, the facilitators shared some of our own related experiences, and the collective story sharing from all sides sparked deeper conversations among the participants. Participants led the group discussions towards the subjects they found the most pressing and raised questions to one another, giving them a voice in the research inquiry. Aligned with the tenets of Sista Circles, these conversations developed organically and were anchored in participant empowerment of one another [73]. A brief example of the dynamics is included in S2 Appendix. Although we did not code transcript excerpts from the research team members during the data analysis process described below, our experiences of engaging in conversation with the participants gave us a nuanced understanding of the data and enabled us to frame the present research in a way that is authentic to how the collective counterstory building unfolded. After the interview participants completed an exit survey (S3 Appendix).

## Reflexive thematic analysis

Reflexive thematic analysis (RTA) is an interpretative qualitative method [82]. We chose this approach for three reasons: 1) We wanted to avoid coding processes that center qualitative reliability; 2) We did not have preexisting ideas about codes or themes before data collection; and 3) We believed our identities as members of the LGBTQ+ community should not be left at the door when analyzing data or synthesizing findings. In order to conduct RTA, one must have four orientations within the data: a theoretically driven approach to analysis (deductive), a focus on understanding the meaning behind the surface of the data (latent), a critical theoretical framing, and a focus on the social construction of meaning. The research engaged the three phases of RTA by doing the following:

Phase 1: Familiarization. Each team member wrote analytical memos after each interview, followed by individual reflection on the transcripts. Group discussions helped synthesize insights from the data.

Phase 2: Coding. An iterative process was used, beginning with *in vivo* coding, followed by team discussions to develop shared definitions of the codes. Group members drew on their own lived experiences as LGBTQ + individuals to inform the coding process. The codes generated in this phase are presented as a codebook in S4 Appendix.

Phase 3: Generating Themes. The team clustered codes with shared meanings and synthesized broader themes that reflected the experiences of trans chemistry graduate students, focusing on how these students navigate their education and the institutional structures around them.

### Reflexivity statements

The first step in RTA is the acknowledgement that the researcher's position, identity, and contribution is an integral part of the analysis process. The "reflexive" component of RTA involves drawing on the researcher's preexisting knowledge and relationships to critically interrogate how our position in the world influences our understanding of the data. For that reason we first present here the reflexivity/positionality statements of the research team:

**Nolan.** Michelle (any/all) is queer and genderqueer. After completing my PhD in chemistry, I left the field and became a chemistry librarian. I have remained active in LGBTQ + chemistry affinity groups and I have served as an LGBTQ + policy advisor in the field. I am a grassroots community organizer and I have lived experience with the research topic. I am a lifelong resident of a trans-hostile US state and I experienced the recent reactionary political shift firsthand while working on this project. As a white academic with US citizenship, Isaac and Paulette held me accountable for ensuring my privileges were not projected on the data during analysis.

**Blythe.** Isaac (they/he) is queer and trans. I am also an inorganic chemist and I was a graduate student at the time of initial data collection and analysis. As such, I am part of the demographic being studied in this work and have first hand knowledge of many of the experiences described by participants. Due to the population size of "out" trans chemistry graduate students in the US and the method of participant recruitment, I know many of the participants and was held accountable for not bringing this external knowledge into the analysis by Michelle and Paulette.

**Vincent-Ruz.** Paulette (she/ella) identifies as a queer latine cis woman of color. I was born and raised in Mexico City. English is my second language, and I have an accent when I speak English. As the only cis member of the research group I acknowledge that I have participated in trans erasure and oppression in my past research related to "Gender and STEM." I was not part of any of the interviews as I wanted to make sure my presence didn't disrupt the safe space created for participants to share their stories. Furthermore, I was held accountable by Michelle and Isaac on not bringing my biased cis perspective or on trying to unconsciously impose a binary analysis in our data.

### Results

As a result of our thematic reflexive analysis, we synthesized five themes from the data. Table 2 summarizes the synthesized themes and includes exemplary excerpts from the data for illustration. Throughout the Results and Discussion sections, direct quotes from the participants are presented in italicized font for clarity. We made an explicit choice to title each theme using the participants own words.

**Table 2. Themes synthesized through reflexive thematic analysis. The themes were titled in the participants' own words (included as a "Titular Quote").**

| Theme | Description | Titular Quote | Example Quote |
|---|---|---|---|
| "I Just Hedged My Bets" | Estimating the risks of being trans in a chemistry doctoral program based on limited information. Making a best bet when choosing a department in which to study. | *"I had a hard time because a lot of the places that I applied, I didn't know anyone there. And like, I hadn't been there, obviously. So I kind of just hedged my bets." -Kayden* | *"I mean, so, for me at least it was, I wouldn't be able to tell for sure if it was a friendly campus. Like policies are great, they're going to get you so far, but it's not like you physically know someone at each of those schools or you feasibly could contact someone at each of those schools." -Jack* |
| "I Feel I Live Two Lives" | Sharing, covering, or compartmentalizing trans identity in chemistry spaces. Negotiating outness in different situations based on assessment of safety. | *"It's very alienating. I feel I live two lives, my queer life out of lab and then my life in lab, which I hate a lot." -Alex* | *I use sie/hir pronouns when possible, but a lot of people aren't familiar or comfortable with them so she/her depending on the day can even feel better. And at work, I use he/him. [...] I haven't said anything about my gender at work. I've started wearing nail polish and growing my hair long and putting bi pride flags in my hair. So, some people have probably guessed some things. -Theo* |
| "Decide Your Own Level of Outness" | Agency (or lack thereof) over one's identity. Control of names, pronouns, and how students are identified in their institutions. Decisions made on students' behalf based on their legal identification. | *"When you fill out an application, you should be able to decide your own level of outness to a given institution. I had some schools that would just straight up put my deadname on a name tag, and I was like, 'that's a no.'" -Nat* | *"The first thing that comes to mind off the top of my head, that almost made me not apply to the school that I am now at [...] is that you could only choose like male or female gender options. [...] I'm like glad I didn't care because I don't think that was representative of most of the school, but I very nearly just did not apply at all." -Cameron* |
| "Static Versus Active Support" | Whether institutional culture and policy is superficially supportive or genuinely meets trans students' needs. Whether institutions prioritize appearances over action. | *"One of the topics that's been on my mind recently is like the concept of "static" versus "active" support. When you go and you look at a lab website or an LGBT Center website, that's static. That's a nice sign to see. But what really mattered to me was evidence of active support." -Nat* | *"I wasn't offered any administrative help, but I have a really excellent PI and she sat down and helped me through it and that was really helpful and supportive. And she sort of just let me CC her on all the emails that I sent just so that she could also just have a set of eyes and made sure that when I sent emails, she explicitly told me, "Make sure that you say that I'm watching, as well." -Farren* |
| "I Am Full Up on Educating Other People" | Empowerment and/or disillusionment arising from self-advocacy. Opportunity cost of invisible and emotional labor associated with trans advocacy. | *"I really only feel like I am full up on educating other people and I'm very tired. It just feels like you're just the token trans person that they want [to advise] on everything." -Farren* | *"I am the tokenized out trans person in my department. [...] I am always asked to sit on panels to attract graduate students because I am a secret, hidden, S-rank diversity choice to put on the panel 'cause they unlocked the trans student who works really hard and blah, blah, blah, blah, blah." -Indigo* |

## Theme 1. "I Just Hedged My Bets"

This first theme encompasses all the factors students took into account when choosing a graduate program and the difficulty of weighing their decisions. Most of the students undertook extensive research about state, local, and institutional LGBTQ+ policies, geographic location, trans community presence, gender-affirming healthcare access, personal contacts, and other LGBTQ+ support systems before they even considered submitting applications to chemistry doctoral programs. The specific variables participants weighed in their decision-making process are included in the S5 Appendix, but ultimately, the participants described having to "hedge bets" on the risks of being trans in a prospective program. For example Jack, a transmasculine person who is "*from a northern place, now living in a southern place*," shared that he had one acceptance offer in a state where legislation limiting trans people's access to public restrooms was under consideration during his admissions process.

> At the time, the state that I was going to be moving to was proposing bathroom bills. And literally the legislation was happening the same like week or two after visitation weekend. And so it was just kind of that awkward time, and then it died in committee a week before I was supposed to start there. So it was just kind of in general an internally tumultuous time. That wasn't going to be something that anyone would be able to answer except for the state government. And so that was just kind of looming in the back of my head. Is this actually going to affect me? -Jack

Complicating the situation, many participants had no avenues through which to ask essential questions for their evaluation process without disclosing their trans identities ("*outing*" themselves) to other people, which caused "*turmoil*" for some participants. Many times, policies about health insurance coverage, restroom access, or campus LGBTQ resource centers were unavailable or unreasonably difficult to locate. Prospective departmental climate could only be assessed through "*academic whisper networks*," where the students relied on "*word of mouth*" endorsements or warnings from their undergraduate mentors, current graduate students in prospective departments, and other personal contacts for advice. The participants described having to read "*between the lines*" to gauge whether a department would be supportive of their trans identities, usually from online information, departmental communications, or campus visits.

The difficulty of finding information in university or departmental websites often pushed students to use location as a proxy to estimate how trans-friendly an institution would be.

> *I had come out a year before and I had not pursued any medical transition options yet. I had looked into it, but being from the rural south, I ran into many roadblocks in trying to get that. So I had not yet started. So one of the big things on my mind was how easily it would be to medically transition. And so I applied to schools, probably somewhat naively, based upon where trans people seemed to be very accepted. So I applied to some schools in [US west coast]. I had some other schools in mind that I decided not to apply to because they were geographically located also in the south or the southeast. I had a school in Texas that I was really interested in, and I did not apply because I did not want to be in Texas.* -Indigo

Moreover, when the research team asked participants whether any prospective departments or individual professors proactively gave indicators of trans support, examples were slim. The only participant who experienced a meaningful show of support was Anna, a trans woman who had already developed a relationship with a future PI whom she trusted to be her advocate. Some participants ($N = 3$) noticed when faculty provided their pronouns in email signatures or Zoom names, which the participants interpreted as a positive sign. Indigo, a trans woman, offered a similar observation when thinking back on her entry to graduate school: for her, the absence of information about diversity, equity, and inclusion (DEI) efforts from a department should have been an unspoken signal of hostility.

> *If I was to give advice to people that are coming into graduate school from what I know, basically, if they don't say anything about it [DEI] at all, then you should stay away. If they say something, then I can't tell you whether or not to stay away or not. [...] But if they don't say anything about the diversity at all, then don't go there.* -Indigo

This resulted in a recurring narrative during the group interviews about how the only way to truly know how "*trans friendly*" a place will be is to live there personally or know another trans person who lives there, neither of which were feasible for most prospective institutions. Hedging bets was a source of distress because ultimately the participants "*wouldn't be able to tell for sure if it was a [trans] friendly campus*" and could not ensure their safety at an institution before committing to attend.

> *I also had a hard time like without – because, a lot of the places that I applied, I didn't know anyone there and like I hadn't been there obviously so I kind of just hedged my bets. [...] When it was in places where I'm like, "Hmm, this area has a reputation of being not friendly," I chose cities where like, it was more likely that, you know, even if there is no one, I could find*

*someone. Yeah, and I also - I remember scrolling through like social media, especially because I was looking at schools that had LGBT centers, if the LGBT Center social media existed and had like visibly not straight/cis people I was like, "ah, ok," it's like there's a certain level of "I'm not going to get beat up here" which is nice. -Kayden*

Some of the participants preemptively assumed that they would not find other trans people within their prospective chemistry departments. Instead, they looked to the greater university campuses and surrounding municipalities for clues about connecting with local trans people in a kind of balancing act.

*One other big thing that I'm considering is the location. […] In the middle of a major metropolitan area, I'm much less concerned about finding a queer environment at the school because there is a major metropolitan area right next door that I could escape to. If it's a university that is located in a small college town of 10,000 people, I'm much more concerned about the school having a queer community, and having those queer support networks there in the university rather than outside the university, if that makes sense. -Eris*

Finally, even after going through all this research and decision making, two participants called their past selves "*naive*" for believing these narratives about location because they did eventually face hostility in their graduate programs, even in "*accepting*" places.

*I think it was a little naive thinking that I can just get away with being in an inclusive city, even if the university itself was trash. -Farren*

## Theme 2. "I Feel I Live Two Lives"

The second theme centers around the experience of participants navigating complicated negotiations regarding the presentation of their trans identities while applying to graduate school and acclimating to graduate programs. Due to internal and external factors outlined here, many participants felt the need to hide or omit aspects of their identities in the admissions process and throughout their graduate education to varying degrees.

*It's very alienating. I feel I live two lives, my queer life out of lab and then my life in lab, which I hate a lot. -Alex*

Participants first navigated whether to disclose their trans identities in the written graduate school application. The data intake consequences of these applications are discussed in depth in Theme 3 ("Decide Your Own Level of Outness"), but the philosophical question regarding whether to authentically present their gender identities in the application was an underlying issue that affected the majority of the participants (all but one who identified as trans at the time of application, $N=7$). Some of the participants did not want to reveal their transness at the written application stage for a variety of reasons, including fear that discrimination from graduate admission committees would place increased scrutiny on their application materials. Eris, a nonbinary transfemme, exemplified these concerns when they felt unspoken pressure to represent herself in a palatable manner for the presumed cisgender evaluators of their application.

*I had to have a very internal conversation with myself, and maybe some other people I know who are my advocates, on whether I should use she/her pronouns or they/them pronouns in my application materials. Specifically, in my letters of recommendation. I was tempted to*

*ask them to change pronouns every other sentence, but I was advised that would probably not look great. So I did not instruct my letter writers to do that. I think I defaulted to she/her pronouns because I didn't want the potential for an old man reading my application to subconsciously or consciously judge my application based on the merits of what pronouns I use. [...] It was a source of incredible turmoil, deciding whether to censor my identity or whether to express my identity because of the potential for impropriety in application judging.* -Eris

Others chose to cover their trans identities entirely, including Kayden, who said they "*talked about things other than being trans*" in their personal statements because "*they were just safer topics*" to disclose. Without knowledge of who would be judging their application materials, Kayden found the risk of discrimination to outweigh their desire for authentic expression.

Alex, a genderfluid transfemme, approached the graduate application as a kind of litmus test for prospective institutions, believing that if she was forthcoming about her trans identity and was still accepted into the program, then those departments must be somewhat welcoming to trans people. The process of doing so was nonetheless painful for Alex because she was personally exploring her understanding of gender and pronoun choices. Finding a way to articulate their identity was challenging, but Alex felt that it was necessary to ensure their department would "*see me as a person*" instead of "*just a chemist on a bench.*" For Alex, being a trans person and being a chemist were inseparable parts of their identity which both needed to be acknowledged.

*The fact that I had to explain my identity in each application was very hard for me. It was like, I had to apply for a school, talk about myself, but I don't know who I am. It was so hard, and trying to choose pronouns, it was a mess. I didn't like that. [...] It did make me feel uncomfortable. It was really bad. But I told myself that I had to be as open as possible just so that I make sure if they pick me, they know who I am already. Which was hard for me to accept it and go through it while writing, but I think it did pay off in the end.* -Alex

Several participants struggled with writing required diversity statements in their applications. Particularly, there was friction with "*Women in STEM*" demographic categorization, with which the participants felt an implicit pressure to align in order to access gendered support networks and to be "*counted*" as a gender diverse member of the student cohort. Kayden, for example, said "*Well, I'm not a woman, but also you really don't have many of me, as a like nonbinary or trans student. So it's like, I guess I'll click that box?*" Kayden wanted to be acknowledged as a gender minority in the applicant pool, but cisnormative categorization of binary men and women excluded them from being recognized. Jack also struggled with articulating a diversity statement, which was an optional portion of their application, and eventually decided to not include a statement in their application at all.

*It was kind of like a "damned if I do, damned if I don't" situation because like if I write this from the perspective of "I'm not applying as out but I will be out on your campus," then like, to fill out the application, I have to be like, "Well I'm furthering diversity by being a woman in STEM" and that's just, I don't want to write that essay. And if I write it from the perspective of being out, I'm like "I'm adding diversity or program by actually being a white guy," like that - there's just, I don't know what you want from me, but I'm still adding diversity if you actually respect me and acknowledge me like as this masculine human being.* -Jack

After starting their graduate programs, students' decisions about whether to share their trans identities in their place of study were deeply influenced by their connection to other trans

people and their immediate social environments, whether affirming, apathetic, or hostile. Connection to other trans people was essential for students to feel like they had a community where they could be their authentic selves. Cameron, a nonbinary person, "*braced*" themselves for "*not being out at work and school*" when starting their graduate studies because they were skeptical about finding any trans peers in their new institution. They were "*pleasantly surprised*" to meet a senior trans student in their department and learn that they were not alone, saying "*there aren't very many of us, but we're not the only trans people in the department.*" The presence of a small trans community within their chemistry department empowered Cameron to come out as nonbinary on campus and ultimately made Cameron satisfied with their choice of school.

Conversely, the absence of fellow trans students left the majority of participants feeling alienated, isolated, and as if they live "*two lives*" inside versus outside the academy. Participants who described themselves as the "*only*" trans graduate students in their departments to their knowledge ($N=7$) experienced acute feelings of isolation and alienation. Indigo, a trans woman who said she is the "*token*" trans person of her department, described feeling constantly othered and without community among her chemistry peers.

> *As someone who's a joint student in two different labs and also basically the only trans person in my department, I feel like I belonged nowhere.* -Indigo

Farren, a trans man, spent "*the first three years of graduate school establishing my queer community that was outside of the department.*" His status as the only out trans person in his program led to heightened personal visibility, but also intense isolation. As a graduate student undergoing transition-related medical care, Farren's experiences blended the lines between typical graduate school challenges (e.g., coursework, research, teaching) and trans-specific challenges (e.g., medical care, transphobic climate). However, he felt that he did not have a platform to discuss his struggles with fellow students because his cisgender peers could not relate to his experiences and responded with discomfort, making him feel that his struggles had no place being discussed in chemistry spaces. He was only able to find support through "*non-university, non-graduate students*" external to his institution.

> *People just avoid me or act like walking on eggshells around me. And that sucks. [...] I don't like feeling like I'm hanging out with a bunch of snowflakes where you just can't say certain things about transness or queerness. There's not really a floor for you to complain about how difficult it is being trans and in grad school. And I feel like that's the way that you bond with other graduate students is you just complain about how hard it is. And when there's no other trans people, it's like... Yeah. It was really funny. In o[rganic] chem[istry], I had to draw all my mechanisms on the board and they were all on the bottom half of the board because I had gotten top surgery and wasn't allowed to raise my arms up above my head. And then if you say stuff like that, cishet people freeze because they just don't know what to say.* -Farren

Nonbinary participants found themselves in the uncomfortable position of needing to either enforce usage of their lived pronouns or allow themselves to be misgendered by faculty and peers, many of whom were ignorant of or resistant to gender neutral pronouns. The cognitive and emotional labor of constantly correcting pronoun usage was "*exhausting to say the least, and potentially dangerous to say the most,*" leading a few participants to stay fully or partially closeted at work and present themselves as their sex assigned at birth. Theo, a genderqueer person, covered hir identity to accommodate other people's discomfort with neopronouns

and never felt comfortable sharing hir trans identity in chemistry settings, instead "*defaulting*" back to hir pronouns assigned at birth.

> *I use sie/hir pronouns when possible, but a lot of people aren't familiar or comfortable with them, so she/her depending on the day can even feel better. And at work, I use he/him. [...] It's hard. There's not the social time to sit down and come out. [...] I'd need some way to meet the other queer people in the department because we can kind of be invisible.* -Theo

While interacting with chemistry faculty, the participants made situational decisions about passing, covering, or authentic presentation based on their perceived level of safety. In a particularly alarming example, Eris experienced a dehumanizing interaction with a potential advisor where the faculty member was attempting to "*clock*" her, meaning that the professor was trying to determine whether Eris was trans and make invasive assessments about her anatomy. In this situation, Eris felt coerced into holding back information about their gender identity and trying to pass as a cisgender woman in order to minimize harassment from a departmental authority. Even reflecting on this situation during the group interview was palpably uncomfortable for Eris.

> *There was a potential PI that I met with who I believe was actively trying to clock me during our interview. She asked leading, borderline questions to try to gather more information about my identity. She was toeing the line of not asking me outright what was in my pants, but she was implying wanting to know why I made certain choices about my identity and my presentation. Um, which is like, you know, that's not, that's not super kosher.* -Eris

Cumulatively, the participants felt pressured to separate their trans identities from their chemistry work and keep their presentation on campus strictly "*scientific*," despite their aversion towards (or even ability to) do so.

## Theme 3. "Decide Your Own Level of Outness"

This third theme relates to the lack of agency trans students have over when and how their lived gender identity is presented to the university, department, advisor and coworkers. We will explore particularly how the set up of university's information systems have a trickle down effect into their personal interactions in their chemistry department. These information systems are particularly important because not all trans students have government identification reflecting their lived names and gender identities. When information systems don't provide space to give "preferred names" students are essentially set up to be misgendered and deadnamed with every first-time interaction they have within an institution.

For example, Anna, a trans woman who was already living under her chosen name before applying to graduate programs, did not have the option to provide her lived name to the university she wound up attending. This violation of her agency led to the disclosure of her deadname to others, assignment of an email address under the wrong name, and confusion about who she was within her chemistry department for the first several months of her doctoral studies. Anna shared that "*everybody was very good about just switching over*" to her lived name in spoken interactions once she settled in, but the impacts of these cisnormative identity management systems were painful and unnecessary. For Anna, being able to provide her lived name upfront would have saved her from the effort of repeatedly coming out to correct others on how she should be addressed.

*For the application process, it would be great if you could put down a lived or preferred name, and how you want that used in contacting you, or contacting your permanent address, anything like that. It would be really great if the university had a system in place that allowed your email to have that name so that everybody in the department isn't confused for a few months. That'd be really great. Yeah, that's the big one for me. I'm very bitter.* -Anna

Other institutions did provide the ability to provide a preferred name (or sometimes a "nickname"), but multiple participants were frustrated to find that this information did not appear to be used. Kayden said that of all the programs they applied to, "*one school used the name that I put as my preferred name, every other school used my legal name*" in communications. Kayden learned after enrolling in their graduate program that online institutional portals "*show other people my deadname.*" They did not realize this discrepancy was widespread across institutional systems until other people began to ask them about their deadname, which made them feel "*miserable times ten.*" Jack also explained that even though he had been given the opportunity to provide his lived name in some applications, the information linked to his academic records, email correspondence, and online learning platforms were inexplicably addressed to his deadname anyway. In characterizing how the application data was used in reality, he called the "preferred name" data meaningless.

*It would be like "Here, input your legal name. And then if you have a preferred first name, you can enter this in another box here. Except everything ever is going to be linked to your legal name, not your preferred name. I don't know why we're even asking you for this, honestly."* -Jack

Information systems also had broader implications for trans student's safety. Two participants had their trans identities disclosed to their recommendation letter writers through automated emails triggered by the application management systems, and two other participants were outed to their families through written mail sent to their permanent addresses. Nat, a nonbinary transfemme, was not out to her undergraduate professors when she submitted applications to graduate programs. Most institutions sent recommendation letter requests listing her legal name, but she was surprised that one institution sent letter requests under her "preferred name," disclosing her trans identity. Nat said that having the ability to tailor how these automated emails were sent out would have saved her from the discomfort of coming out to a professor before she was ready.

*When you fill out an application, you should be able to decide your own level of outness to a given institution. I had some schools that would just straight up put my deadname on a name tag, and I was like, "that's a no."* -Nat

Another consequence of this lack of foresight presented during students' experiences when interviewing at the institution. Campus visit accommodations were predetermined for the participants based on intake data without the students' input. Legal gender marker data were used to make decisions on the participants' behalf regarding gendered spaces (e.g., hotel roommates, bathrooms). Even when institutions asked for "preferred name information", three participants said that upon arriving for a department visit, they were handed printed name tags with their deadnames displayed. Jack was extremely uncomfortable wearing a name tag with his deadname, but did not want to endanger himself by correcting people onsite. Instead, he chose to literally cover his nametag.

"*I just kind of flipped it around for the entire weekend, but if I saw my name out, I would flip it back. [...] I wasn't going to come out to strangers if it wasn't going to be a meaningful connection, but I also really was in a space where I could not be referred to by the other [dead] name.*" - Jack

At the department level, participants' legal name data were internally circulated to introduce the applicants to faculty members without consulting with the students beforehand. Several participants experienced direct confrontations about their lived names by faculty, dehumanizing them and outing them to their peers. In a meeting with fellow prospective students and the chemistry department chair at one institution, Kayden said the chair was initially confused about their name. When Kayden specified how they would like to be addressed, the chair insisted "*I don't see why I can't just use your legal name*" and continued to do so. Nat experienced a similar confrontation in front of an audience, during which Nat felt pressured to cover her trans identity and acquiesce to cisnormative expectations of professionalism.

> At one institution, I had a professor straight up ask me at a table full of people why I used my current name instead of my legal name. And that was a very awkward question to answer. A part of me wanted to be like "Oh, well I use that name because I'm transitioning," and part of me wanted to be like, "Well, let me not burn that professional bridge." -Nat

Finally, when trans students tried to update their institutional records, several participants ran into dead ends where departmental representatives did not know the relevant procedures, if procedures existed at all, and did not know an appropriate campus referral. Information databases at institutions were often not connected to each other, requiring students to interact with multiple campus offices to update their names or gender markers. This impasse left trans students to fend for themselves without support. When Anna sought to update her information, "*nobody knew what to do, nobody had done it before, and they didn't actually have a good process in place for doing it.*" She had to figure out how to get these changes made herself, which required outing herself to multiple people across three different campus offices. Kayden initially sought help from their department, but instead was pointed through a series of referrals that ultimately led to no satisfactory method for updating their information.

> The only thing I really want or need from my department is knowing who to point me out to ask these things because when I was trying to figure it out I got pointed to like three different people who all like pointed to three more different people. [...] And thankfully I found the right person, but their answer was, "Oh we know this is a problem and we're going to fix it in the future. Is it okay if we wait until then?" And it's like, do I have to? Could you jerry-rig a fix for me now? -Kayden

Farren, a trans man, was able to get his government identification updated to reflect his lived name. However, his deadname was still in the university information systems and was never rectified, despite numerous attempts to correct his personal information. He said that his university's internal information system was easily capable of changing last names, but not first names, leading to two different employee personnel files being associated with his student record. The complete failure of the institution's identity management systems led to egregious, recurring financial harm, including missed paychecks and double tuition charges.

> It's a total fucking dumpster fire. And it's led to things like me getting overcharged. [...] My tuition remission is late every term because they have a mix-up, even though it [his lived

*name] is my legal name. [...] There have been multiple times where I didn't get paid on time. I've been double charged for terms as if I was two people because there's two names associated with my account and they can't figure out why that is. Just nothing was easy about it. Nothing. Four years later, I just got an $8,000 bill in the mail because the tuition remission was for my deadname, not for my legal name.* -Farren

The only participants who experienced a smooth transition regarding identity data at their new institutions were those who had completed legal identification updates prior to applying to graduate programs or who were using a variation of their legal name that was perceived as a nickname (e.g., a truncated version).

## Theme 4. "Static Versus Active Support"

The fourth theme addresses the ways in which institutions and professors "talk the talk" but cannot "walk the walk." When retelling their experiences participants often made distinctions on whether a department was "*passively*" inclusive (i.e., not overtly hostile) or "*actively*" supportive of trans people. In the participants' words, these "*static*" or "*passively supportive*" climates often amounted to apathy or "*wanting to look presentable*" without implementing real change.

> *One of the topics that's been on my mind recently is like the concept of "static" versus "active" support. When you go and you look at a lab website or an LGBT Center website, that's static. That's a nice sign to see. But what really mattered to me was evidence of active support.* -Nat

The general atmosphere that many of the participants described in their departments can best be illustrated in the following quote from Cameron, a nonbinary person.

> *No, like there wasn't anything where it was like, "you're not going to be okay with this," but there also wasn't anything where it was like, "you will not misgender me." And I think that every professor that I've talked to so far in my department has at some point misgendered me. [...] I think if you ask them, they'd be like, "Yes I care," but they don't care enough to, you know, actually do it. Actually pay attention to the words that they're saying. [...] No one is aggressively bad. But also, no one here that I've met, none of the professors, are aggressively, you know, good either.* -Cameron

After being chronically misgendered, Cameron believed that faculty members only "*cared enough*" about trans inclusion to avoid accusations of bigotry, compromising their trust in the department. The unspoken social contract of the department seemed to be that their transness was tolerable so long as it didn't require extra effort from faculty to accommodate.

During the group discussions, the participants raised examples of failures to support trans needs in university policy and working conditions which they believed conflicted institutional DEI promises, including the following issues: graduate student health insurance benefits that do not cover transition related care, absence of gender inclusive restrooms on campus, inability to take time off for court appearances related to government document updates, unavailability of medical leave for transition related care, and lack of recourse for students who are chronically misgendered or deadnamed. Indigo explicitly named these frustrations while talking about the lack of gender neutral restrooms in the chemistry buildings on campus. After approaching the departmental leadership about bathroom access multiple times, Indigo concluded that her department chair "*wants to appear like he gives a shit, but [...] he just wants the status quo to remain exactly the way it is.*"

The students implicitly understood that trans inclusivity is a matter of doing and not just saying. For example, Farren observed that his department made regular statements about how fostering a diverse group of scientists will improve the pursuit of science. Rather than this rhetoric being accompanied by real action to make the department friendlier to marginalized people, he noticed that the statements were simply used as evidence of a good climate itself. For Farren, the true goal seemed to be cultivating an image of an inclusive space, rather than meaningfully making the department a better place to work while trans.

> *If you truly believe that your scientific community will be better and you will do better science with more diverse people, you won't hesitate to make sure that all of these things are inclusive and go out of your way to do the research to make sure that what you're doing is inclusive. I think that my department went from "not giving any fucks" to "wanting to look presentable" because of the way that the culture of universities is changing. More of [an attitude of] "I don't want to offend anybody." And I think that that's maybe another step of, "Good. You're thinking that not everything fits everybody." But I think what would be even more meaningful is that if people actually bought into realizing that we're only going to solve the problems of the world if the representation of humanity is reflected in departments.* -Farren

The participants also brought up instances where institutional DEI policy did not offer specific enough guidance to support trans students, leaving decisions up to the participants' supervisors. A major example was access to transition-related healthcare: even after carefully considering health insurance benefits and legal landscapes, participants still encountered roadblocks where chemistry departmental culture inhibited the students' ability to pursue transition-related care. As Nat pointed out, the time consuming nature of transition is not well understood by cisgender people, even those who consider themselves allies. Accessing gender affirming medical care and government document updates are long term processes that require doctors appointments, legal consultations, court appearances, and recovery time, often over the course of multiple years and involving many dimensions of oppressive gatekeeping [83]. Faculty who have higher expectations of their graduate students' research productivity were less likely to approve students taking time off work for medical procedures, regardless of their ostensibly trans inclusive personal sentiments. In Nat's assessment, a university providing health insurance that covers transition-related care without ensuring graduate students could use medical leave to actually access that care was another example of a false commitment to trans inclusivity. She said "*I need not just the resources to be able to transition, but I need to feel like I have the time to go and use them, otherwise it doesn't mean anything.*"

Nat strategically sought out faculty members who would "*actively*" support her as her advisor, eventually choosing a PI who supported her physical and mental wellbeing.

> *I came out to him explicitly and I mentioned, you know, about coordinating time off if I ever wanted to have anything medical done. And he was like, "Oh, that's cool. I know all the appropriate channels for that. So if you want to take me off, you're totally welcome to." [...] A lot of it was me trying to almost justify myself and justify my transness in front of this person. And me being like "Hey, this is not going to be a burden on your lab. This is a positive thing for me. And even if I do have to be out [from work], [...] maybe I can take up some computational work while I'm away from the lab." But you know, in the end he was just "No, it's fine." Good work-life balance person.* -Nat

Nat realized in retrospect that when she approached her future advisor about these questions, she was discussing her medical needs as if they were a potential liability to her PI's research

group. Fortunately, Nat's advisor interrupted this thought process and ameliorated her concerns, encouraging her to take time off for her health. Anna had a similar experience: her university and department did not have any proactive policies in place to address trans health-care, but she was individually encouraged by her advisors to take ample time off to recover from surgeries. Both Nat and Anna described that "*support felt more on an individual basis, rather than institutional one*," and still described their overall departments as merely "*passively supportive.*"

The trans affirming actions of individual people made a huge difference for the participants, even through simple gestures like pointing out gender neutral restroom locations. This individual support is a key consideration in the mentorship of trans students throughout their graduate education. However, individual supervisors being solely responsible for supporting trans students' needs is untenable and inconsistent: many of the participants did not have any champions they could entrust for help while navigating "*the social, the legal, and the medical bureaucracies*" that shape trans life in the academy. The participants also noted that this mode of individualized help was offered because specific faculty or staff members knew (or inferred) them to be trans, making the participants concerned that the same guidance was not provided to the larger student cohort. That is, other students who do not disclose their trans identities or who are currently gender questioning would not be made aware of the same resources.

## Theme 5. "I Am Full Up on Educating Other People"

Our final theme reflects how the lack of institutional policy regarding support of trans students creates a vacuum that students need to feel with their own invisible labor. All of the participants ($N = 10$) engaged in some level of self or community advocacy in varying forms. Advocating for the trans community within their chemistry departments was a source of empowerment when the students were acting of their own volition. A common thread was that the participants were not just looking to protect themselves, but felt a sense of responsibility to improve trans visibility and the material working conditions in their departments for future trans students who would follow them. Many of the participants mentioned common practices to normalize trans existence in chemistry spaces, such as hanging pride flags in office or laboratory spaces, posting affirming flyers in gendered restrooms, including pronouns in online correspondence, and participating in prospective student recruitment events.

> *I'm really aggressive about including my pronouns in places now. And so they're in my email signature. They're in my zoom stuff. When we had physical name tags, they were on my physical name tag. They're in all of those places. If you come to my office, I have little safe space stickers and like flags and stuff. [...] I've had enough conversations one-on-one with different professors at the university that they've all started adding stuff like pronouns to their signatures, so now the department chair has pronouns in his email signature. And like, little stuff like that has been just really how I end up pushing.* -Jack

Theo was not out as genderqueer in hir workplace but sie found that addressing pronoun sharing in the abstract influenced hir department, saying "*since I started putting my pronouns in emails, after I email professors, I've noticed that they start adding their pronouns to emails too.*"

However, some of the openly trans participants were also pressured to provide unpaid labor for their departments in the forms of serving on committees, providing LGBTQ+ training to their peers, representing the department on panels, and consulting on policies.

The cumulative toll of their advocacy work was apparent to the research team, manifesting as burnout and disillusionment.

> I *really only feel like I am full up on educating other people and I'm very tired. It just feels like you're just the token trans person that they want [to advise] on everything. And even the things that don't even have to do with LGBTQ issues or the social culture of the department.* -Farren.

This was also reflected by Eris, a nonbinary transfemme, who was in the process of interviewing with several graduate programs and had not yet committed to an institution. At her undergraduate institution, Eris had already been deeply involved in organizing campaigns related to trans inclusivity on campus. Now that they were moving into a graduate program, they were dreading the amount of labor they might need to provide regarding the same issues at a new institution.

> *A lot of the stress that I have about grad school is not necessarily whether I'll feel included, but more about how much shit am I going to have to wade through to get where I want to be. Rather than, how much of that work is already done for me. Because I've done this work [trans advocacy] before. But I don't necessarily want to do it again, if I can help myself. I will do it again. [...] Just to make sure that trans people survive, both me and anyone else in the program behind me. How much wading through that bureaucracy am I going to have to do? How much unpaid labor am I going to have to do, in that regard? Rather than doing my job as a graduate student and trying to get the hell out. Because if I'm devoting 20 hours a week to trying to change the institutional processes that oppress us, that's 20 hours a week that I'm not in lab doing research or sleeping.* -Eris

Alex, a genderfluid transfemme in their first year of graduate school, also had previous experience advocating for gender inclusive restroom access in her undergraduate institution. Alex felt frustrated "*going back to gendered bathrooms*" in graduate school because the chemistry buildings had no gender inclusive restroom options, despite the fact that "*there's a lot of trans nonbinary students in our department and the numbers are going up.*" For Alex, the lack of restroom availability "*makes no sense*" and they felt called to fight for change on behalf of current and future nonbinary students. Indigo, who was near the end of her graduate program, was also "*pulled between a bathroom situation and writing my thesis*" and felt compelled to "*fight*" the department chair on behalf of her trans peers. Despite the demands of her educational obligations, getting involved as an advocate in this situation felt obligatory due to her both her own conscience and the expectation from her peers that she would speak out.

Farren, who came out as a trans man during his graduate studies, took it upon himself to start educating others in his department about how to respect trans people through conversations, flyers, and training sessions. These acts were initially a source of empowerment for Farren who was coming into his gender identity and sharing willingly with his peers. However, once Farren had developed a reputation as an activist, he was inundated with people approaching him to ask questions about the trans experience. His status as the only out trans person in his department came with a high pressure to not only perform exhaustive emotional labor for his cisgender peers and mentors, but an expectation that he would speak for all trans people collectively on trans issues.

> I *had a graduate student who saw me in the hallway, and this is just maybe a month ago, saw me in the hallway and then proceeded to be like, "Hey. Back home, there was a*

*restaurant that had a thing where they decided to say something very anti-trans. And my parents go to that restaurant. And I want to have a conversation with them about how that's not appropriate. What do you think I should say?" And I was [thinking] like, "I really am just trying to go run an NMR [nuclear magnetic resonance experiment]. Can you please just leave me alone?" But I obviously stood there for five minutes and explained to him and I looked at the Facebook page and the post and that sort of thing. And, I don't know, it just doesn't feel like I have the option to be like, "No, I'm not going to help you." -Farren.*

Farren also was tapped to volunteer on multiple departmental committees and policy groups, service that is typically performed by faculty members as part of their professional responsibilities rather than graduate students as unpaid volunteers. Beyond advising on LGBTQ + topics, he was also asked to serve on unrelated committees (such as chemical safety) as a "*diverse voice.*" Farren was frustrated because "*I am a white, cis-passing, masculine person. I do not count as your diversity token.*" He also felt pressured to excel in these service appointments because he was perceived as the monolithic voice of the trans community, bestowing a high level of both expectation and scrutiny on his performance.

*I think that I tend to overwork myself in sort of those volunteer positions because I'm also the tokenized trans person. And so then I do feel like I'm held to a higher standard. I mean, at least in terms of performing in a leadership role or performing in a volunteer role, that I definitely feel a lot of external and probably internal pressure to not perform badly because I'm out and visible and do advocacy work. -Farren*

The participants were also skeptical about the extractive use of their identities for the benefit of their department's reputation. Being counted as a "*diversity token*" rang hollow when there were no accompanying improvements in the trans exclusionary culture of the department. For example, Indigo was repeatedly asked to represent the chemistry department on panel discussions and to participate in graduate student recruitment. She said that she was "*the tokenized out trans person in my department*" and that the department thought of her as a "*secret, hidden, S-rank [top rank] diversity choice*" who could be used to "*attract*" prospective graduate students. For Indigo, using her existence as evidence of an inclusive department was frustrating. The high burden of time and emotional labor required for these events detracted from writing her dissertation and progressing towards graduation, and ultimately, speaking at these events did not improve her daily working conditions. These speaking engagements only benefited her department, rather than herself or the trans community. In the end, Indigo was simply exhausted.

*Shit's hard. It's just hard. It's not easy. And people should just recognize that. And then recognize that I'm putting in more work for the same amount of productivity as your cis, white, female and male students. -Indigo*

## Discussion

### Theme 1. Calculated risks in choosing a chemistry graduate program

The results from this theme highlight how the complex decision-making process for trans students is unique when applying to graduate school. Research has shown that cisgender women and racially marginalized students are more likely to account for social factors when choosing a graduate program [84–86]. However, our participants' took into account factors that were consequential in ways unique to the trans experience because life-altering

conditions (e.g., medical autonomy, queer community networks, safety) were weaved with the factors all new graduate students must consider (e.g., university resources, future research agenda). Our Theme 1 findings were similar to Goldberg's recent exploration of trans graduate students' decision making processes while choosing a graduate school, where participants described making tradeoffs between academic factors and personal factors when evaluating programs [21].

Without information about trans support in a prospective department, the participants were forced to rely on cultural narratives about location to make their best judgment calls (e.g., liberal US states are more accepting, large cities are more diverse, the US South is more trans antagonistic). Almost all of the participants ($N=9$) discussed location as a high or highest priority factor in comparing graduate programs. Location was largely discussed as a proxy measure for other priorities (e.g., political climate, bodily safety, existence of local trans communities) and must be understood against a backdrop of increasingly hostile legislation targeting trans people in the US [87]. That is, students were not only hedging bets on campus climate, but also on whether the future legislative environment would constrict their freedoms during graduate school [53,88–90].

The participants also felt resigned to the fact that there would be limited or non-existent trans community inside prospective chemistry departments, leading them to prioritize locations where they hoped to find a critical mass of trans people external to the university. These tradeoffs explain why in contrast to prevailing advice on choosing a graduate program [91], a minority ($N=4$) of the ten participants discussed their future research agenda as a factor that influenced decisions. When chemistry research areas were raised, it was always in combination with other weighted factors (e.g., location, mentor suggestion) and research agenda was not considered the highest priority for any participant. We observed that future research was only discussed in a way that prioritized supportive and trans inclusive doctoral advisors, rather than research interest in isolation. The relatively low attention given to doctoral research agenda demonstrated that for trans students, non-academic factors are often more important for choosing a graduate program than academic factors.

Ultimately, the interviewed students were trying to gauge whether they could survive and thrive as trans people at an institution based on limited or unspoken information. Whether a deliberate choice by the institution or not, the absence of information about trans needs sent the participants a message about the school's priorities and who belongs on campus. Echoing the results from Goldberg and coauthors, our participants expressed that for trans students, "there is no 'perfect school'" [21].

## Theme 2. Compartmentalization and negotiating outness

As shown in the results of this theme, weighing whether to come out as trans, with whom to share their identities, and when to have these coming out conversations were questions to approach with utmost care given the perceived inappropriateness of discussing personal identity in STEM spaces. This tension was especially salient for the nonbinary participants, who experience coming out as a constant process of conversation with no true arrival at "*being out*" because "*every time that you meet someone new, you have to come out and put yourself at risk.*" Participants described a need to separate much of their identities as trans people from their identities as scientists, leading to an isolated experience that required them to "*live two lives.*"

Before even submitting applications, many participants were making decisions that would ultimately impact how they navigated matriculation and the early years of graduate school by necessarily choosing aspects of their identities to disclose in their personal statements and other application materials based on fear of discrimination from admissions committees,

similar to sentiments reported by trans jobseekers [92]. Eris's thought process and the decision to ultimately restrict her pronoun usage in application materials to be palatable to assumed cisgender application reviewers reflect what scholars have termed "covering," in which members of an oppressed group feel obligated to downplay what is perceived as a negative trait in order to emulate or assimilate into dominant norms, behaviors, and expectations [93]. As Nicolazzo has described, trans people may feel "compelled" to partially or fully cover their transness in response to "perceived or real threat" in an environment, responding to trans oppression with self-censorship [1]. Eris wanted to authentically express their identity through varying usage of *she* and *they* pronouns, but because the nature of nonbinary identity is unintelligible to many cisgender people [94,95], she felt obligated to compromise on using one set of gendered pronouns against her wishes. On the other hand, some participants utilized the application as a method of pre-screening institutions. This strategy was similarly observed by Goldberg and coauthors, who found that some trans graduate students used being out in their written statements as "a way to effectively 'weed out' potentially invalidating environments" [21]. The "*Women in STEM*" demographic question was particularly consequential to navigate when it was associated with gendered economic or academic opportunities (e.g., fellowships, awards). The students understood that these initiatives were designed to broaden participation in chemistry, but because their own gender identities were not acknowledged, they experienced trepidation (or as Nat described, "*trans imposter syndrome*") about whether they were viable applicants.

Compartmentalization of transness outside of chemistry spaces echoes back to the "don't ask, don't tell" culture of STEM, inflicting harmful psychological consequences [56] and inhibiting epistemological border crossing. This reality is important to confront: studies about trans people in the greater workforce have demonstrated reduced job and life satisfaction for trans people who feel uncomfortable presenting their full, authentic selves at work [96]. Connection to other queer and trans students was influential on the participants in deciding whether to share their trans identities on campus, which resonates with reported literature about how LGBTQ + STEM students build spaces for themselves and each other to resist cis-heteronormative STEM learning environments [66]. Without overstating the significance of "coming out" as the end goal for all trans people, we want to stress that chemistry departments must work towards creating a climate where trans students feel they have the option to present their authentic selves in STEM spaces. Shifting chemistry culture is a long term process, but we would be remiss if we did not point out that the absolute minimum necessary is an end to faculty members questioning graduate students about their bodies.

## Theme 3. Agency of identity and personal information disclosure

The lack of robust structure in application websites and university information systems created a situation where trans students were denied the opportunity to determine how their data would be shared with individuals from their new graduate departments, personal circles, and undergraduate institutions. The absence of choice exposed their private information without their consent, jeopardizing their relationships and financial security. For these students, having the foreknowledge of how their data would be used and the ability to choose how their names were presented to others would have kept agency over their identity in their own hands. These issues with personal data for trans people are not unique to university settings [97–101], but had major implications for the participants during graduate school.

Specifically, the students' stories show how institutional systems designed with only cisgender people in mind, including these identity management systems for academic and employment records, make trans students vulnerable. The participants did not have any methods to

protect themselves and maintain agency over their identities, leading to misgendering and deadnaming in their departments. If epistemological border crossing is the goal, then these systems are setting students up for failure by enforcing that their trans selves will be incompatible with their lives as scientists. Given that government document updates are not always possible or feasible, trans students often rely on self-description of their lived names and genders. Cisnormative information systems must be rebuilt with trans needs in mind.

Finally, the lack of a culture of respect for trans identities at the departmental level opened the door for professors to behave in ways that humiliated students presenting their lived selves. There is little we can say as authors to express how unprofessional and dehumanizing chemistry professors were to the participants. Addressing these entrenched attitudes about trans people goes beyond improving information intake forms and educating people in our departments. The participants' experiences illustrate the need for a massive cultural change in STEM to provide safety and an affirming climate for trans scientists. Whitley and coauthors have argued that chronic misgendering of graduate students in the academy is itself pedagogical because "it teaches participants and bystanders to dismiss transgender people's identity and self-designated pronouns" as part of the professional socialization in their fields [20]. Misgendering and reassignment of deadnames to the participants was a similar tool of disempowerment in these chemistry departments. Faculty members and universities did not consider the participants to be the sole information authority concerning their names and instead revoked the participants' agency over their identities, reinforcing the trans-antagonistic culture of chemistry and teaching bystander peers to do the same.

## Theme 4. Superficial versus substantive trans inclusivity

Our results demonstrate that participants were looking for departments that intentionally cultivated systems and communities that enable trans students to thrive. However, they saw few examples of institutional work specifically addressing the needs of trans students. For example, although one participant responded to a question about whether they had faced any clear discrimination towards their nonbinary identity by saying "*no*," they conversely did not experience any affirming treatment either. The liminal experience of moving through a department that neither explicitly excluded nor actively included them sent the message that trans people are simply not a priority. These participants' observations were similar to reports about the culture of engineering departments towards queer identities, where students described their college as neither "welcoming" nor "anti-welcoming," but "silent" [61].

The participants' critiques closely align with feminist theorist Sara Ahmed's argument that higher education institutions engage in repetitively stating their commitments to DEI as the outcome itself, rather than as a promise to guide continuous practice [102]. Ahmed coined this type of rhetorical commitment to DEI as a "non-performative speech act," where *naming* takes the place of *doing* and ultimately fails to bring about what is named. Grappling with these contradictions between institutional DEI messaging and material reality highlighted the broken trust between participants and their departments: the students witnessed little interest from their departments in confronting systemic barriers to trans existence in chemistry. Some participants were disillusioned with the prospect of future improvement. The compounding silence and absence of supportive policies were further examples of how the overall institutional and STEM culture hinders participant's ability to engage in epistemological border crossing because their needs as trans people were not met or acknowledged in their chemistry environments. Failure to genuinely support trans students was also apparent when transition-related procedures are considered optional, unnecessary, or distractions from work, rather than essential medical care by faculty. The lack of institutional policy leaving students' fates to the discretion of individual advisors must be readdressed.

## Theme 5. Invisible labor and the price of advocacy

In the absence of institutional support, the participants felt called to advocate for their own inclusion in their chemistry departments through a range of actions. Addressing trans topics in a general sense (e.g., stickers, flags, pronouns in email signature) was one way that participants who were not out on campus felt that they could shift the departmental culture without disclosing their trans identities, although we want to acknowledge that this type advocacy (i.e., for nonspecific others rather than for oneself) has still been associated with the negative psychological effects of trans identity covering in the workplace [96]. Outspoken advocates faced consequences: trans students searching for respect and dignity in their departments eventually morphed their existence into a resource from which cisgender people could learn, demanding their continual patience and draining their time and cognitive resources away from their graduate studies.

The participants' experiences reflect what Berenstain termed epistemic exploitation, where privileged people "compel marginalized persons to educate them about the nature of their oppression" through uncompensated and unacknowledged labor [103]. Numerous studies have explored the impact of emotional and invisible labor on people of color and cisgender women in academia, ranging from students to faculty members [104–106]. This body of literature highlights the disproportionate expectations placed on marginalized academics to match the productivity and "excellence" of their privileged peers while also shouldering the additional burden of emotional and invisible labor, revealing outcomes such as burnout, disillusionment, and missed career opportunities. Our research team did not encounter similar studies in the literature specifically focusing on the experiences of trans people, but the narratives shared by our participants resonated closely despite trans identity sometimes being considered a "concealable" identity.

Finally, it's crucial to recognize that any form of advocacy, whether at the micro or macro level, could potentially lead to repercussions for students because it influences how departments and institutions perceive them. When students are positioned as instigators of change, the relationship dynamic between students and their universities shifts from being primarily focused on education and labor to being inherently political or even adversarial. Those who point out oppressive conditions in their institutions become perceived as the problem themselves because they disrupt the cultivated image of their university [107,108]. While not explicitly discussed by the participants, the research team was also concerned about repercussions in the field for activist students who might be mischaracterized as disruptive by chemists with power over their future careers.

## Limitations

A critical consideration for the research team was developing this study from a place of trust and safety for the participants. As such, the call for study participation was shared through LGBTQ+ affinity group email lists, which then further circulated through word of mouth sharing. This mode of closed network sharing certainly left some voices out, particularly for potential participants who were not out or not already connected with LGBTQ+ chemistry communities. We maintain, however, that this does not invalidate the results of this study, as trust and safety are paramount to study queer communities. Our recruitment method centered on prior development of trust with participants and anonymity. Limited demographic data were collected on participants to minimize risks to the participants being identified, meaning that compounding factors to student experiences in graduate school (e.g., racial identity, disability, socioeconomic status) were not available to the research team for analysis. Our participant recruitment method also focused on enrolled or enrolling graduate students,

which inherently leaves out the stories and perspectives of trans chemists who ultimately did not pursue graduate studies or left the field entirely. We also wish to emphasize that the direct examples of trans oppression raised by this study are not applicable to all trans people, as the experiences of trans folks are not homogenous. The purpose of presenting these counterstories is to draw attention to institutional and cultural barriers our participants faced in chemistry graduate programs, rather than to overgeneralize or project their individual stories. We aim to raise the voices of these students so that scholars and practitioners can interrogate their own institutions for similar barriers and implement change.

## Conclusions

This research highlights the unique and overlooked challenges trans graduate students face within academic institutions. From the beginning of their academic journey, these students were forced to navigate complex tradeoffs between academic goals and personal safety, often without reliable information about the level of support or inclusivity they can expect. This uncertainty adds an additional layer of stress to an already challenging process, with students unable to fully anticipate the conditions of support or oppression they may encounter in their programs. Once enrolled, trans students frequently experienced a compartmentalization of their identities, feeling that their transness is invisible or irrelevant in academic spaces. This lack of recognition often results in isolation and the need to rely on informal networks or individual mentors, rather than formalized institutional support. Connection to other trans people was essential for students to feel like they had a community where they could be their authentic selves. Conversely, the absence of fellow trans students left participants feeling alienated, isolated, and as if they live "*two lives*" inside versus outside the academy. Additionally, the harmful practices of deadnaming and misgendering within institutional systems erode students' control over presenting their identities, underscoring the urgent need for reforms that empower trans students as the authorities of their personal information. Finally, our findings demonstrate that trans students are burdened with invisible labor and advocacy, often taking on the role of fighting for their own inclusion within academic spaces. These efforts place additional strain on students and can create adversarial relationships with their institutions. To foster genuinely inclusive environments, universities must rethink how they manage student data, provide trans-informed mentorship, and actively create policies and cultures that support the specific needs of trans students.

Since the completion of our group interviews, there has been a rapid intensification in trans-antagonistic rhetoric and politics across the United States and beyond. Several of the fears participants raised in the abstract (e.g., potential "*bathroom bills*") have since become a material reality with which trans people must contend. Based on what we learned doing this research, we believe the increasingly hostile political environment has already had far-reaching effects on the choices current trans students will make regarding attending graduate school and presenting their trans identities at their place of study. We end this article with an urgent call to action for members of academic communities to move the findings of our work into practice and create real institutional change.

## Supporting information

**S1 Appendix. Interview Questions.**
(DOCX)

**S2 Appendix. Exchange Transcript.**
(DOCX)

**S3 Appendix. Exit Survey.**
(DOCX)

**S4 Appendix. Codebook.**
(DOCX)

**S5 Appendix. Hedged Bets Theme.**
(DOCX)

## Acknowledgments

We would like to thank all of the research participants for sharing their stories and experiences with us. IMB would like to thank their Ph.D. advisor, Prof. Melanie S. Sanford, for flexibility and time to carry out this research in tandem with their dissertation work.

## Author contributions

**Conceptualization:** Michelle M. Nolan, Isaac M. Blythe.

**Data curation:** Michelle M. Nolan, Paulette Vincent-Ruz.

**Formal analysis:** Michelle M. Nolan, Isaac M. Blythe, Paulette Vincent-Ruz.

**Investigation:** Michelle M. Nolan, Isaac M. Blythe.

**Methodology:** Paulette Vincent-Ruz.

**Project administration:** Michelle M. Nolan.

**Writing – original draft:** Michelle M. Nolan, Paulette Vincent-Ruz.

**Writing – review & editing:** Michelle M. Nolan, Isaac M. Blythe, Paulette Vincent-Ruz.

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
