## [Decision Letter · Decision Letter 0]

13 Aug 2024

PONE-D-24-22765“Behind closed doors: The untold challenges of transgender and nonbinary graduate students in chemistry”PLOS ONE

Dear Dr. Vincent-Ruz,

Thank you for submitting your manuscript to PLOS ONE. After careful consideration, we feel that it has merit but does not fully meet PLOS ONE’s publication criteria as it currently stands. Therefore, we invite you to submit a revised version of the manuscript that addresses the points raised during the review process.

 Major revisions to the manuscript are required as outlined by the peer reviewers below. I would like to highlight the - in my and the reviewers view - excessive length of the submission. I would strongly suggest to consider ways to reduce the length of the paper.

We look forward to receiving your revised manuscript.

Kind regards,

Daniel Demant, PhD, MPH, GradCertHEd, BAppSocSc

Academic Editor

PLOS ONE

Additional Editor Comments (if provided):

Reviewers' comments:

Reviewer's Responses to Questions

**Comments to the Author**

1. Is the manuscript technically sound, and do the data support the conclusions?

Reviewer #1: Yes

Reviewer #2: Yes

2. Has the statistical analysis been performed appropriately and rigorously? 

Reviewer #1: N/A

Reviewer #2: N/A

3. Have the authors made all data underlying the findings in their manuscript fully available?

Reviewer #1: No

Reviewer #2: Yes

4. Is the manuscript presented in an intelligible fashion and written in standard English?

Reviewer #1: Yes

Reviewer #2: Yes

5. Review Comments to the Author

Reviewer #1: The reviewer is grateful for the opportunity to review this manuscript. The researchers worked on an important issue that merits publication in PLOS One. However, a number of revisions should be considered before the manuscript is deemed ready for publication. The reviewer's suggestions are as follow:

Title

As much as it pains the reviewer to limit on the authors' creativity, the title currently feels like a book title rather than a factual report which, in its essence, is the format to which scientific journal articles should adhere. The phrase "Behind closed doors" is figurative and should be removed to avoid misunderstanding. The challenges of sexual and gender minority students are being told with this article, thus the term "untold" is now invalid and should be removed. The qualitative findings using the Sista Circles methodology is innovative and should be highlighted. Thus, the reviewer wishes to suggest that the authors consider changing the title to:

"The challenges of transgender and nonbinary graduate students in chemistry: a qualitative study with Sista Circles methodology"

Please kindly note that this suggestion is optional.

Abstract

Please make the abstract in the structured format with Background, Methods, Results, Conclusion

INTRODUCTION

Paragraph 1

* Please add 1-2 sentences to provide details or examples of marginalization and oppression in STEM that are unique to trans students. This will help readers to are unfamiliar with the topic to understand the context of the study outcome.

* Please also consider making an argument (1-2 sentences) on how empowerment of LGBTQ+ people can benefit STEM. This can help stakeholders to see that empowerment will not only benefit the LGBTQ+ community from the social justice perspective, but also benefit STEM promotion overall.

* The reviewer finds the last sentence of the first paragraph to be confusing. Please elaborate on the potential danger and the dynamic hostile environments.

Paragraph 2

* Please integrate the content of this paragraph with the first paragraph.

* The reviewer wishes to caution the authors regarding the use of determinative language (such as "Without community input and building trust, we create unreliable data that are incapable of empowering LGBTQ+ people" placed after a sentence regarding the need for qualitative data to contextualize information). If possible, please make the rationale more focused.

* The reviewer was confused by the sudden appearance of the issue of omitting BIPOC people from quantitative studies in STEM promotion

* Overall, the second paragraph needs extensive revision

Third paragraph

* The term "critical transition" was confusing. Critical with regard to what? Career decisions? Please revise the opening sentence.

* The authors should consider whether to include examples of the "distinct factors" that trans STEM students consider as part of the hypothesis statement.

STEM Culture and Epistemological Border Crossing sub-section

* First paragraph: Perhaps the authors should start by contextualizing STEM disciplines culture in general, and then mention the marginalization fo trans students. Please also mention the potential effect of cisheteronormative oppression on the mental health and performance of trans students.

* Second paragraph: Please consider integrating the content of this paragraph with the previous paragraph. Please also consider making the literature view more concise.

* Third paragraph: Please consider adding an argument on how a better understanding of epistemological crossing among trans students in STEM (particularly chemistry) can be of use for stakeholders in LGBTQ+ rights as well as those in STEM education

METHODOLOGY

Design and Participant Recruitment

* Please consider separating this section into three sections: 1) study design and setting; 2) study participants eligibility and exclusion criteria; 3) participants recruitment. The study design and setting sub-section should contain clear statements that data collection was done via the internet.

* The authors should also consider adding a separate sub-section titled "Study Instrument" with details on how the semi-structured interview protocol and guidelines were developed for the Sista Circles sessions. If possible, please include details regarding the validation and pilot-testing processes (or lack thereof).

Group Interview Procedure

* Please include information on how the investigators collected data during the Zoom call (i.e., whether the investigators made a recording of the Zoom session with the participants' consent and the subsequent transcription of the recording, if any).

Reflexivity Statements

* The reviewer appreciates the researchers' willingness to fully disclose the reflexive component of RTA on the manuscript for transparency and to share with the readers their preexisting knowledge, relationships, and experiences. However, as personal experiences are not replicable and contextual, the reviewer wonders whether the authors should consider making the reflexivity statements as part of the supplementary section and replace the section with the statement somewhat like the following:

"The researchers performed a reflexive component of RTA among themselves to hold one another accounting for not projecting own privileges on the study data during analyses, the details of which can be found in the Supplementary Material section."

In that regard, the reviewer is a cisgender heteronormative male epidemiologist from South East Asia who has worked on LGBTQA+ health disparities in an extractive manner. The reviewer acknowledges that his comments and suggestions can be consider as an erasure of safe spaces for the authors. Thus the above-mentioned suggestion should be considered as optional.

RESULTS AND DISCUSSION

* The reviewer wonders whether the two section should be kept as two separate sections, with the RESULTS section containing description of identified themes and the existing quotes, whereas the DISCUSSION section containing the broader discussion regarding the study findings and their implications.

CONCLUSION

* The reviewer feels that the current version of the Conclusion section reads more like a part of the DISCUSSION section or the rationale in the INTRODUCTION section. Please consider making extensive revisions.

Implications for Research & Implications for Policy

* The reviewer wishes to suggest that the content of these sections should only be those based on the research findings.

Reviewer #2: Thank you for the opportunity to read this manuscript. It provides an interesting and important contribution regarding the challenges for trans and nonbinary graduate students in chemistry. I have a few suggestions to offer in terms of development of the manuscript, which I hope the authors will find useful.

It wasn’t clear in the introduction that the focus of the paper was on graduate chemistry students. Adding this to the sentence, ‘in this project we chose to focus on the critical transition moment…” would help to orient the reader.

In terms of the methodology, I wanted further information on Sista Circles and the process of co-constructing knowledge through storytelling, and how and why this is supportive and builds community. The description of the analysis could be more concise.

The results themes were clear. However, the findings section was overly long and risks losing reader interest. That said, the authors provided long stretches of information with few exemplars from their data, and instead move to a broader discussion of their findings. I would suggest more exemplar quotes, more concise explanations, and creating a discussion section.

Finally, the conclusions and links to policy and practice are interesting, but not well linked to the wider international literature, and again rather detailed.

This is an interesting paper, and I hope the comments are useful to the authors.

6. PLOS authors have the option to publish the peer review history of their article (what does this mean? ). If published, this will include your full peer review and any attached files.

**Do you want your identity to be public for this peer review?** For information about this choice, including consent withdrawal, please see our Privacy Policy .

Reviewer #1: No

Reviewer #2: No

---

## [Author Response · Author response to Decision Letter 1]

18 Oct 2024

Dear Editor and Reviewers,

We would like to express our sincere gratitude for your thoughtful and constructive feedback on our manuscript. Your insights and suggestions have been invaluable in improving the quality and clarity of our work. In this letter, we have carefully addressed each of your comments and concerns. We hope that our revisions meet your expectations and enhance the manuscript accordingly. We appreciate the time and care that both reviewers have taken to provide these constructive comments, and we trust the revisions address your concerns. We look forward to your feedback on our revised manuscript.

Thank you again for your time and effort in reviewing our work.

Sincerely,

Paulette Vincent-Ruz, Ph.D. (she/ella)

Assistant Professor in Chemistry Education Research

Dept of Chemistry and Biochemistry

New Mexico State University

P.O. Box 30001, MSC 3C

1175 North Horseshoe Dr.

Bldg 187, Room 103

Las Cruces, NM 88003-8001

(575)-646-3554

Reviewer 1

Comment:

The title feels more like a book title and should be factual to align with the scientific format. The figurative phrase "Behind closed doors" should be removed, and the term "untold" is now invalid. The reviewer suggests the following title:

"The challenges of transgender and nonbinary graduate students in chemistry: a qualitative study with Sista Circles methodology."

Response:

Thank you for this suggestion. After careful discussion, we agree that a more descriptive title is appropriate. We have revised the title to:

"The challenges of transgender and nonbinary graduate students in chemistry: a qualitative study on trans identity, science culture, and institutional support using reflexive thematic analysis."

This title strikes a balance between scientific precision and conveying the essence of our work.

Comment:

Please structure the abstract with Background, Methods, Results, and Conclusion.

Response:

As far as we know, PLOS ONE does not require a structured abstract. However, the current abstract includes all necessary elements: background, methods, results, and conclusion. If the editor prefers a structured format, we will happily revise it accordingly.

INTRODUCTION

Comment:

In Paragraph 1, please add details/examples of marginalization and oppression unique to trans students in STEM and make an argument on how LGBTQ+ empowerment benefits STEM. The last sentence is unclear and should be elaborated or removed.

Response:

We have revised the introduction by including examples of the unique marginalization trans students face in STEM. Additionally, we expanded on the benefits of LGBTQ+ empowerment for STEM, emphasizing that it is a matter of human dignity beyond its advantages for the field. The unclear sentence has been removed for clarity. The entire introduction has been restructured for conciseness and clarity.

Comment:

Please integrate the second paragraph with the first and avoid determinative language. Clarify the sudden mention of BIPOC exclusion in quantitative STEM studies.

Response:

The second paragraph has been integrated with the first, and we have revised the determinative language for precision. The reference to BIPOC exclusion has been clarified to ensure the flow of argument is coherent.

Comment:

The term "critical transition" in Paragraph 3 is confusing. Provide examples of "distinct factors" considered by trans STEM students.

Response:

We have replaced the term "critical transition" with more precise language, highlighting the pivotal nature of graduate applications in career trajectories. We have also provided examples of the distinct factors considered by trans STEM students.

STEM CULTURE AND EPISTEMOLOGICAL BORDER CROSSING

Comment:

Please start by contextualizing STEM culture in general before addressing the marginalization of trans students. Integrate the second paragraph with the first, make the literature more concise, and add an argument on the relevance of epistemological crossing for stakeholders.

Response:

We appreciate these suggestions. The section now begins with a broader context of STEM culture, followed by a discussion of how it marginalizes trans students. We have included citations on the mental health of trans graduate students and integrated the second paragraph for a more concise literature review. We removed the discussion on epistemological border crossing since it was not reflected in the results, but we retained the focus on how STEM culture shapes the scientific experiences of trans students.

METHODOLOGY

Comment:

Consider reorganizing the methodology into distinct subsections: 1) Study Design and Setting, 2) Participant Eligibility, 3) Participant Recruitment. Additionally, add a "Study Instrument" section with details on interview protocols and validation.

Response:

We have reorganized the methodology into the suggested subsections for clarity. Regarding the interview instrument, we emphasize that our methodology relies on counter-storytelling, which allows participants to guide the discussion. Therefore, validation in the traditional sense was less relevant. We have added a paragraph explaining the participant-led nature of our interviews.

Comment:

Consider moving reflexivity statements to the supplementary section, with a brief statement in the manuscript itself.

Response:

We respectfully decline this suggestion. Reflexivity statements are integral to reflexive thematic analysis, as they account for the researchers' influence on data interpretation. They are a key component of our chosen methodology, and moving them would compromise the integrity of our analysis. Therefore, we have retained them in the main manuscript.

RESULTS AND DISCUSSION

Comment:

Separate the Results and Discussion sections for clarity.

Response:

We understand the reviewer's concern. However, separating these sections would expand the manuscript length and disrupt the synthesis of participant narratives with our analysis. Instead, we have condensed the combined Results and Discussion section for clarity and brevity, ensuring the themes remain central to the participants' stories.

CONCLUSION

Comment:

The Conclusion reads more like a continuation of the Discussion. Please revise extensively.

Response:

We have rewritten the Conclusion to provide a succinct summary of our findings, clearly distinct from the Discussion and Introduction sections. The new Conclusion focuses solely on the insights drawn directly from our research.

IMPLICATIONS FOR RESEARCH & POLICY

Comment:

Limit the content of these sections to findings based on the research.

Response:

We have removed these sections, as per your suggestion.

Reviewer 2

Comment:

It wasn’t clear in the introduction that the focus is on graduate chemistry students.

Response:

Thank you for pointing this out. We have revised the introduction to explicitly state that our focus is on graduate chemistry students.

Comment:

Provide further details on Sista Circles and their role in co-constructing knowledge.

Response:

We have expanded the methodology section to explain Sista Circles in greater detail, focusing on how this method supports community building and empowers participants through co-constructed knowledge. The process of counter-storytelling is now described more thoroughly to illustrate its impact.

Comment:

The findings section is overly long with insufficient exemplar quotes. Consider more concise explanations and adding more quotes.

Response:

We have revised the findings section to be more concise, incorporating additional exemplar quotes to better illustrate the key themes while maintaining the clarity of our broader discussion.

---

## [Decision Letter · Decision Letter 1]

16 Dec 2024

PONE-D-24-22765R1The challenges of transgender and nonbinary graduate students in chemistry: a qualitative study on trans identity, science culture, and institutional support using reflexive thematic analysisPLOS ONE

Dear Dr. Vincent-Ruz,

Thank you for submitting your manuscript to PLOS ONE. After careful consideration, we feel that it has merit but does not fully meet PLOS ONE’s publication criteria as it currently stands. Therefore, we invite you to submit a revised version of the manuscript that addresses the points raised during the review process.

I would like to apologise for the length it took to process the revisions. It has become increasingly challenging to find reviewers in order to provide authors with high-quality feedback. While the reviewer feedback is extensive, the reviewers agree that the manuscript makes a valuable contribution to understanding the experiences of transgender and nonbinary graduate students in chemistry. However, they highlight areas requiring revision to enhance its impact. Both reviewers emphasise the need for a more comprehensive and up-to-date review of the literature, particularly incorporating broader discussions on transness in academia, STEM contexts, and intersectional perspectives. One aspect would be the provision of a clearer theoretical framework, expanded discussion of systemic inequities in academic spaces, and improved integration of the results with the existing literature. Both reviewers also highlight the importance of addressing intersectionality.

Methodological concerns are raised, particularly regarding the Sista Circles methodology. Reviewer 2 suggests elaborating on how this approach aligns with the study’s trans-specific focus and how the co-created knowledge through discussions was handled analytically, given the commitment to reflexive thematic analysis. These revisions, alongside minor points about the reflexivity statements and discussion of systemic policies, are essential to address the constructive feedback and further refine this significant contribution to the field.

We look forward to receiving your revised manuscript.

Kind regards,

Daniel Demant, PhD, MPH, GradCertHEd, BAppSocSc

Academic Editor

PLOS ONE

Reviewers' comments:

Reviewer's Responses to Questions

**Comments to the Author**

1. If the authors have adequately addressed your comments raised in a previous round of review and you feel that this manuscript is now acceptable for publication, you may indicate that here to bypass the “Comments to the Author” section, enter your conflict of interest statement in the “Confidential to Editor” section, and submit your "Accept" recommendation.

Reviewer #3: (No Response)

Reviewer #4: (No Response)

2. Is the manuscript technically sound, and do the data support the conclusions?

Reviewer #3: Partly

Reviewer #4: Yes

3. Has the statistical analysis been performed appropriately and rigorously? 

Reviewer #3: Yes

Reviewer #4: Yes

4. Have the authors made all data underlying the findings in their manuscript fully available?

Reviewer #3: No

Reviewer #4: Yes

5. Is the manuscript presented in an intelligible fashion and written in standard English?

Reviewer #3: Yes

Reviewer #4: Yes

6. Review Comments to the Author

Reviewer #3: COMMENTS TO AUTHORS:

The manuscript titled “The challenges of transgender and nonbinary graduate students in Chemistry: a qualitative study on trans identity, science culture, and institutional support using reflexive thematic analysis” (PONE-D-24-22765R1) presents a qualitative study on the nuanced experiences of transgender students navigating chemistry PhD programs. This study can be a good starting point for research on the lived experiences of transgender, nonbinary, two-spirit, and gender-expansive students within the context of higher education. These populations navigate significantly distinct and multifaceted challenges compared to their cisgender peers, encompassing social, academic, and institutional dimensions. By delving into their unique perspectives, this research can illuminate systemic inequities, identify barriers to inclusion and success, and contribute to developing evidence-based strategies to foster a more equitable and affirming educational environment for gender-diverse students.

The literature reviewed by the authors is coherent with the topic, but it would benefit from broader and more recent literature on transness and academia to enhance the manuscript. I believe that the relatively limited depth of the literature review, combined with the lack of a clear separation between the results and discussion sections, somewhat detracts from the potential scientific impact of the work.

However, I encourage the authors to revise and resubmit the manuscript after significant revisions so that the readers can fully understand it. Indeed, I think there are some limitations in the manuscript’s current version. I hope the authors will consider my comments helpful in improving the paper and pursuing this interesting line of research.

Abstract:

Line 31. Although a structured abstract could not be required, I would suggest adding the number of respondents, mean age, and gender identity as participants represent a principal focus of your study.

Introduction

1. Pag. 2, line 47. I suggest considering the use of “trans*” to keep the focus on capturing a variety of gender identities and expressions for individuals whose gender identity or expression is different from their assigned sex at birth (e.g., see Tebbe et al., 2014. Revised and abbreviated forms of the Genderism and Transphobia Scale: Tools for assessing anti-trans* prejudice) although not all of them identify as trans due to the variability of nonbinary gender (e.g., Rosati et al., 2022. Non-binary clients’ experiences of psychotherapy: Uncomfortable and affirmative approaches).

2. Pag. 2, line 60. I suggest adding “men” after “cisgender and heterosexual”.

3. Pag. 2, line 65. I would suggest expanding the theoretical framework to include studies on STEM in a broader context:

- Cech, E. A., & Waidzunas, T. J. (2011). Navigating the heteronormativity of engineering: The experiences of lesbian, gay, and bisexual students. Engineering Studies, 3(1), 1–24;

- Patridge, E. V., Barthelemy, R., & Rankin, S. R. (2014). Factors impacting the academic climate for LGBQ STEM faculty. Journal of Women and Minorities in Science and Engineering, 20(1), 75–98.

4. Pag. 2, line 72. Given the context in which this study is situated, I would suggest considering that this is not merely a matter of humanity but also an institutional duty and a fundamental right to education first and to employment later.

5. Pag. 3, line 77. I would suggest delving deeper and including additional studies on disparities in academia before addressing those in doctoral programs (e.g. Russell et al., 2016). Mental health in lesbian, gay, bisexual, and transgender (LGBT) youth).

6. Pag. 3, line 102. It would be better to state that they do not fully capture the experiences of trans individuals.

7. Pag. 3, line 114. After “decisions” I suggest adding “and act as barriers to access for these individuals”.

8. Pag. 4, line 116. The literature already demonstrates the impact of cis-heteronormative systems in academic contexts, including in STEM fields (e.g., Hughes, B. E. (2018). Coming out in STEM: Factors affecting retention of sexual minority STEM students). Please revise and enhance the literature review and the theoretical rationale of the study, with a particular focus on situating this research within the broader academic discourse and articulating its specific contributions to the existing body of knowledge. This involves critically examining current studies, identifying gaps or underexplored areas in the field, and demonstrating how this research addresses these gaps. Emphasizing the study's originality and relevance will reinforce its value to scholars and practitioners in the field.

9. Pag. 4, line 119. Considering the intersectional approach of the paper, I would take into account not only race as an additional layer of potential discrimination but also other factors, such as disability.

Methods:

1. Pag. 4, line 131. I would reformulate in “the act of sharing stories serves both as a means of data collection and as…”.

2. Pag. 4, line 137. I would rephrase this section, as the expression “chosen pseudonyms” is unclear. It's uncertain whether it refers to the participants' chosen names as trans individuals or to names explicitly chosen for the study.

3. Pag 5, line 156. It would be more fluent eliding “Importantly”.

4. Pag 5, line 158. I think this part could be made more fluent by omitting the statement from “Acknowledging” to “participant”.

5. Pag 6, line 178. It would be appropriate to justify why the participant chose a pseudonym.

6. Pag 6, line 195. It is stated that the general steps of the RTA are being outlined, yet the specific process of the research is then discussed.

Results:

1. Pag 8, line 256. I recommend italicizing all participant quotations to enhance clarity, even if they contain just a single word.

2. Pag 9, line 295. The number of participants should be capitalized (N=3). This comment also applies to similar situations (e.g., page 11, line 353).

3. Pag 9, line 296. In this section and several subsequent points, it appears that the authors' viewpoints and comments on the research process have been included. While these are interesting, presenting them in a separate section would be more appropriate. I would also like to point out the absence of a section dedicated to the discussion, which seems to have hindered the connection between the results and the existing literature.

4. Pag 12, line 405.I would suggest deepening this result by adding more literature in support:

- Lorusso et al., (2024). Navigating the gap: Unveiling the hidden minority stressors faced by trans and nonbinary clients in gender-affirming pathways. International Journal of Transgender Health, 1–21.

5. Pag 14, line 479. It would be helpful to link this result to the existing literature, particularly regarding social support and transgender individuals.

6. Pag 14, line 502. It would be helpful to deepen this result by referencing various sources in support, particularly regarding the psychological consequences of the described phenomenon (e.g. Pollitt, A. M., Ioverno, S., Russell, S. T., Li, G., Grossman, A. H. (2021). Predictors and mental health benefits of chosen name use among transgender youth. Youth & Society, 53(2), 320–341).

7. Pag 16, line 548. It would be highly beneficial to undertake a more comprehensive exploration of the existing literature on the impact of institutionalized discrimination, power dynamics, and microaggressions within institutional contexts. This deeper analysis should integrate diverse and robust scholarly sources to substantiate the discussion and provide a nuanced understanding of these phenomena. Specifically, the review should examine how systemic biases and entrenched power hierarchies manifest in organizational structures, policies, and practices, perpetuating inequities. Furthermore, it is crucial to explore the pervasive role of microaggressions—subtle, often unintentional discriminatory behaviors—that cumulatively affect the psychological well-being, sense of belonging, and overall success of marginalized individuals within these settings.

8. Pag 16, line 584- It would be helpful to deepen this result by referencing recognizing the chosen name (e.g., Russell et al.,2018). Chosen name use is linked to reduced depressive symptoms, suicidal ideation, and suicidal behavior among transgender youth.

9. Extracts do not accompany some parts of the results. I would suggest connecting the results with relevant extracts better. A good practice would include an extract or a reference whenever a new thematic aspect is introduced.

10. Including a table illustrating the articulation of themes would be highly beneficial, as it would help readers better understand the thematic analysis you have conducted.

Conclusions

It would be interesting to include the perspective of a systemic implementation of policies within academic contexts. One example could be introducing training programs for staff, such as Safe Zone. For further insight on the topic, see:

- Evans, N. J. (2002). The impact of an LGBT safe zone project on campus climate.Journal of College Student Development, 43(4), 522–539.

- Finkel, M. J., Storaasli, R. D., Bandele, A., & Schaefer, V. (2003). Diversity training in graduate school: An exploratory evaluation of the safe zone project. Professional Psychology: Research and Practice, 34(5), 555.

Reviewer #4: Please see attached review. We support this article for publication with minor revisions as outlined in the attached review document.

7. PLOS authors have the option to publish the peer review history of their article (what does this mean? ). If published, this will include your full peer review and any attached files.

**Do you want your identity to be public for this peer review?** For information about this choice, including consent withdrawal, please see our Privacy Policy .

Reviewer #3: No

Reviewer #4: **Yes: ** Wayne Martino

---

## [Author Response · Author response to Decision Letter 2]

15 Feb 2025

As the responses required a lot of information we have attached a file with our response to the reviewers to make it easier for them to track every issue.

---

## [Editor Report · Decision Letter 2]

20 Feb 2025

The challenges of transgender and nonbinary graduate students in chemistry: a qualitative study on trans identity, science culture, and institutional support using reflexive thematic analysis

PONE-D-24-22765R2

Dear Dr. Vincent-Ruz,

We’re pleased to inform you that your manuscript has been judged scientifically suitable for publication and will be formally accepted for publication once it meets all outstanding technical requirements.

Kind regards,

Daniel Demant, PhD, MPH, GradCertHEd, BAppSocSc

Academic Editor

PLOS ONE
---

## [Editor Report · Acceptance letter]

PONE-D-24-22765R2

PLOS ONE

Dear Dr. Vincent-Ruz,

I'm pleased to inform you that your manuscript has been deemed suitable for publication in PLOS ONE. Congratulations! Your manuscript is now being handed over to our production team.

Kind regards,

on behalf of

Dr. Daniel Demant

Academic Editor

PLOS ONE